# Chemical Constituents of the Leaves of *Peltophorum pterocarpum* and Their Bioactivity

**DOI:** 10.3390/molecules24020240

**Published:** 2019-01-10

**Authors:** Yue-Chiun Li, Ping-Chung Kuo, Mei-Lin Yang, Tzu-Yu Chen, Tsong-Long Hwang, Chih-Chao Chiang, Tran Dinh Thang, Nguyen Ngoc Tuan, Jason T.C. Tzen

**Affiliations:** 1Graduate Institute of Biotechnology, National Chung-Hsing University, Taichung 402, Taiwan; ycli0126@gmail.com; 2School of Pharmacy, College of Medicine, National Cheng Kung University, Tainan 701, Taiwan; 3Department of Biotechnology, National Formosa University, Yunlin 632, Taiwan; L3891104@nckualumni.org.tw (M.-L.Y.); black4635@gmail.com (T.-Y.C.); 4Graduate Institute of Natural Products, College of Medicine, Chang Gung University, Taoyuan 333, Taiwan; htl@mail.cgu.edu.tw; 5Research Center for Industry of Human Ecology, Research Center for Chinese Herbal Medicine, and Graduate Institute of Health Industry Technology, Chang Gung University of Science and Technology, Taoyuan 333, Taiwan; 6Department of Anesthesiology, Chang Gung Memorial Hospital, Taoyuan 333, Taiwan; 7Graduate Institute of Clinical Medical Sciences, College of Medicine, Chang Gung University, Taoyuan 333, Taiwan; moonlight0604@hotmail.com; 8School of Chemistry, Biology and Environment, Vinh University, Vinh City 43159, Vietnam; thangtd@vinhuni.edu.vn; 9NTT Institute of High Technology, Nguyen Tat Thanh University, Ho Chi Minh City 72820, Vietnam; 10Institute of Biotechnology and Food Technology, Industrial University of Ho Chi Minh City, Ho Chi Minh City 71408, Vietnam; nguyenngoctuan@iuh.edu.vn

**Keywords:** Fabaceae, sesquiterpenoid, superoxide anion generation, elastase release

## Abstract

Two new sesquiterpenoids peltopterins A and B (compounds **1** and **2**) and fifty-two known compounds were isolated from the methanol extract of *P. pterocarpum* and their chemical structures were established through spectroscopic and mass spectrometric analyses. The isolates **40**, **43**, **44**, **47**, **48**, **51** and **52** exhibited potential inhibitory effects of superoxide anion generation or elastase release.

## 1. Introduction

*Peltophorum pterocarpum* (DC.) Backer *ex* K. Heyne (Fabaceae) is a deciduous tree originated from the tropical regions, ex. Sri Lanka, the Andamans, the Malay Peninsula and North Australia [1]. Traditionally, its flowers are used for slowing intestinal diseases and childbirth pain, treating muscle sprains, bruises, swelling and pain [2]. Roots and barks are also used to cure abdominal colic, joint and back pain, and ascites [3]. Reports on *Peltophorum* species have described antibacterial [4,5], antifungal [6,7], antivirus [8,9], antioxidant [10], antitumor [10,11], deworming [12,13], hypoglycemic [2,14], cardiotonic [15], hepatoprotective [16] and leukoagglutinating bioactivities [17]. However, there are only a few studies related to the chemical composition of the *Peltophorum* species. A preliminary examination showed that the methanol extract and fractions of leaves of *P. pterocarpum* displayed significant superoxide anion and elastase inhibition at 10 μg/mL (Table 1). Therefore, we sought to purify the constituents from the leaf extract and examine the anti-inflammatory potential of the isolated compounds to identify new anti-inflammatory leads from natural sources. In this study the chemical profiles of leaves of *P. pterocarpum* were comprehensively investigated and a total of fifty-four compounds were identified. Among these, two new sesquiterpenoids **1** and **2** were characterized and the structures were established by spectroscopic and spectrometric analyses. In addition, the purified compounds were examined for their superoxide anion and elastase inhibitory effects.

## 2. Results and Discussion

### 2.1. Isoaltion and Identification

Air-dried and powdered leaves of *P. pterocarpum* were refluxed with methanol, and the combined extracts were concentrated in vacuo to produce a brownish syrup. This syrup was suspended into water and partitioned with chloroform to afford a chloroform layer and a water soluble fraction respectively. After isolation using a combination of continuous conventional chromatographic techniques, two new compounds, named peltopterins A (**1**) and B (**2**), were isolated and their structures established by nuclear magnetic resonance (NMR) and mass spectrometric analyses. Moreover, fifty-two known compounds, including six sesquiterepnoids: rel-5-(3*S*,8*S*-dihydroxy-1*R*,5*S*-dimethyl-7-oxa-6-oxobicyclo[3,2,1]-oct-8-yl)-3-methyl-2*Z*,4*E*-pentadienoic acid (**3**) [18], 3,6-dihydroxy-5,6-dihydro-β-ionol (**4**) [19], (−)-boscialin (**5**) [20], (3*S*,5*R*,6*R*,7*E*,9*S*)-3,5,6,9-tetrahydroxy-7-megastigmene (**6**) [21], 2,6,6-trimethyl-4-oxo-2-cyclohexene-1-acetic acid (**7**) [22], (3*S*,5*R*,6*S*,9*S*)-megastigmane-3,9,13-triol (**8**) [23]; nine benzenoids, *p*-hydroxybenzoic acid (**9**) [24], isovanillic acid (**10**) [25], *trans*-methyl p-coumarate (**11**) [24], *trans*-ferulic acid (**12**) [26], methyl ferulate (**13**) [27], benzoic acid (**14**) [28], vanillic acid (**15**) [29], syringic acid (**16**) [30], sodium salicylate (**17**) [31]; two coumarins: scopoletin (**18**) [32], scopolin (**19**) [33]; two lignans: dihydrodehydrodiconiferyl alcohol (**20**) [34], (7′*S*,8′*R*)-7′,8′-dihydro-8′-hydroxymethyl-3-hydroxy-7′-(4′-hydroxy-3′-methoxyphenyl)-1-benzofuranpropanol 9′-*O*-β-d-glucoside (**21**) [18]; one alkaloid: 4(1*H*)-quinolinone (**22**) [35]; one diterpene: 3(17)-phytene 1,2-diol (**23**) [36]; thirteen steroids: a mixture of β-sitosterol (**24**) and stigmasterol (**25**) [37,38], a mixture of 6β-hydroxystigmast-4-en-3-one (**26**) and 6β-hydroxystigmasta-4,22-dien-3-one (**27**) [39,40], a mixture of 7-ketositosterol (**28**) and 3β-hydroxystigmasta-5,22-dien-7-one (**29**) [41,42], a mixture of stigmast-4-en-3-one (**30**) and stigmast-4,22-dien-3-one (**31**) [43,44], β-sitosteryl-3-*O*-β-d-glucoside (**32**) [45], ergosterol peroxide (**33**) [46], ergosta-4,6,8(14),22-tetraen-3-one (**34**) [47], 9,11-dehydroergosterol peroxide (**35**) [48], 20-hydroxy-ecdysone (**36**) [49]; six triterpenes: friedelin (**37**) [50], lupenone (**38**) [51], 24,25-dihydrocimicifugeuol (**39**) [52], cyclotirucanenone (**40**) [53], cycloart-25-ene-3β,24-diol (**41**) [54], cycloeucalenol (**42**) [55]; and twelve flavonoids: kaempferol 3-*O*-α-l-rhamnoside (**43**) [56], quercetin 3-*O*-α-l-rhamnoside (**44**) [57], kaempferol 3-*O*-β-d-glucoside (**45**) [58], kaempferol 3-rutinoside (**46**) [59], quercetin 3-*O*-β-d-glucoside (**47**) [57], quercetin 3-*O*-α-l-arabinofuranoside (**48**) [60], kaempferol 3-*O*-[α-l-rhamno-pyranosyl(1→6)]-β-d-galactopyranoside (**49**) [61], kaempferol 3-*O*-[α-l-rhamnopyranosyl(1→3)]-β-d-glucopyranoside (**50**) [62], quercetin 3-*O*-[α-l-rhamnopyranosyl(1→3)]-β-d-glucopyranoside (**51**) [63], quercetin 3-*O*-[α-l-rhamnopyranosyl(1→2)]-β-d-xylopyranoside (**52**) [64], kaempferol 3-*O*-[α-l-rhamnopyranosyl(1→2)]-β-d-xylopyranoside (**53**) [65], kaempferol 3-*O*-[α-l-rhamnopyranosyl-(1→2)-α-l-rhamnopyranosyl(1→6)]-β-d-galactopyranoside (**54**) [66], respectively, were characterized by comparison of their physical and spectroscopic data with those published previously.

### 2.2. Structural Determination of ***1*** and ***2***

Compound **1** was obtained as white powder with m.p. 116–118 °C and [α]_D_^25^ = −73. The molecular formula was determined as C_11_H_18_O_3_ by a sodium adduct ion peak at *m*/*z* 221.1150 in high resolution electrospray ionization mass spectrometry (HR-ESI-MS) analysis. The infrared (IR) absorption bands at 3430 and 1709 cm^−^^1^ corresponded with the presence of a hydroxy and a carbonyl groups, respectively. In its ^1^H-NMR analysis, there were proton signals for two methyl singlets at δ 0.87 (3H, CH_3_-11) and 0.95 (3H, CH_3_-10), one methylene group at δ 2.20 (1H, dd, *J* = 18.1, 12.2 Hz, H-7) and 2.68 (1H, dd, *J* = 18.1, 5.6 Hz, H-7), two oxymethylene protons at δ 3.88 (1H, m, H-9a) and 4.34 (1H, dd, *J* = 11.1, 4.8 Hz, H-9b), one oxymethine at δ 3.88 (1H, m, H-3), and two methines at δ 1.21 (1H, dd, *J* = 12.1, 6.7 Hz, H-2) and 0.89 (1H, m, H-4). The ^13^C-NMR and distortionless enhancement by polarization transfer (DEPT) spectra exhibited a ester carbonyl signal at δ 171.2 (C-8), two methyl carbons at δ 20.2 (C-11) and 29.2 (C-10), four methylene carbons at δ 30.8 (C-7), 36.8 (C-4), 50.0 (C-2) and 73.8 (C-9), three methines at δ 32.3 (C-5), 44.2 (C-6) and 65.8 (C-3), and a quaternary carbon at δ 34.0 (C-1). The correlation spectroscopy (COSY) correlations of H-2/H-3/H-4, H-5/H-6/H-7, and H-9/H-5 suggested the presence of the partial structures -CH_2_CH(OH)CH_2_-, -CH_2_CHCH-, and -CHCH_2_-, respectively. The ^2^*J*- and ^3^*J*-correlations from H-2 to C-4, C-6, C-10, and C-11; from H-3 to C-5; from H-6 to C-5 and C-9; from H-7 to C-1, C-5, and C-8; and from H-9 to C-4, C-5, and C-8, respectively, could be observed in the heteronuclear multiple bond correlation (HMBC) spectrum (Figure 1). According to these spectral analyses and the index of hydrogen deficiency (IHD = 3), it indicated the presence of two rings and a carbonyl group in **1** and these analytical results constructed the planar structure of **1** (Figure 1). Furthermore, the coupling pattern of full width at half maximum (FWHM) of H-3 (12.3 Hz) revealed its axial orientation. The nuclear overhauser effect spectroscopy (NOESY) showed NOE correlations between H-3 and H-5 but no correlations were observed between H-5 and H-6. These experimental data established the relative stereochemistry configuration of **1** as shown (Figure 1) and assigned the trivial name peltopterin A.

Peltopterin B (**2**) was assigned a molecular formula of C_13_H_20_O_4_ from HR-ESI-MS analysis. The ultraviolet (UV) absorption maxima at 232 nm and the IR absorption bands at 3414 and 1677 cm^−^^1^ indicated the occurrence of hydroxyl and conjugated carbonyl groups. The ^1^H-NMR spectrum of **2** revealed signals for one vinyl proton at δ 5.84 (1H, s, H-7); four methyl singlets at δ 1.15 (3H, CH_3_-12), 1.37 (3H, CH_3_-11), 1.42 (3H, CH_3_-13) and 2.17 (3H, CH_3_-10); one oxymethine at δ 4.33 (1H, dddd, *J* = 11.5, 11.5, 4.0, 4.0 Hz, H-3); and two methylene groups at δ 1.36 (1H, m, H-2_ax_), 1.43 (1H, m, H-4_ax_), 1.98 (1H, ddd, *J* = 11.5, 4.0, 2.5 Hz, H-2_eq_), and 2.29 (1H, ddd, *J* = 11.5, 4.0, 2.5 Hz, H-4_eq_). The ^13^C-NMR spectrum also displayed two carbonyl signals at δ 198.5 (C-9) and 209.8 (C-8), three methyl carbons at δ 26.5 (C-10), 29.2 (C-11), and 31.1 (C-13), two methylene carbons at δ 49.1 (C-2) and 48.8 (C-4), two methines at δ 63.4 (C-3) and 100.9 (C-7), and four quaternary carbons at δ 36.3 (C-1), 72.5 (C-5), 118.6 (C-6), and 31.8 (C-12), respectively. The ^2^*J*- and ^3^*J*-HMBC correlations from H-2 to C-3, C-6, C-11, and C-12; from H-4 to C-3, and C-6; from H-7 to C-1, C-5, C-6 and C-9; from CH_3_-10 to C-9; from CH_3_-11 to C-1; from CH_3_-12 to C-1 and C-6; and from CH_3_-13 to C-4 and C-6, respectively, evidenced the planar structure of **2** as (3,5-dihydroxy-1,1,5-trimethylcyclohexylidene)butan-8,9-dione (Figure 1). The NOE correlation between H-3 and CH_3_-13 (see Appendix A) determined the relative stereochemistry at C-5 (Figure 1) and the structure **2** was characterized accordingly. However, the absolute configurations of the two new compounds remained to be determined. Among the purified flavonoid glycosides, compounds **51** and **52** had been reported without NMR spectral data recorded in MeOH-*d*_4_ [63,64]. In the present study, these compounds were identified through 1D and 2D NMR spectroscopic analysis and the fully assigned NMR data in MeOH-*d*_4_ were listed in Section 3.

### 2.3. Anti-inflammatory Activity

Among these isolates, numerous compounds were selected to be evaluated for the superoxide anion generation and elastase release inhibition by human neutrophils in response to *N*-formyl-l-methionyl-phenylalanine/cytochalasin B (fMLP/CB) (Table 2). 

The results indicated that **43**, **44**, **47**, **48**, **51** and **52** show a significant inhibition of superoxide anion generation, with the inhibitory percentages ranged from 42.3 ± 4.3 to 48.5 ± 1.0% at 10 μM. In addition, **40**, **43**, **47**, **48** and **52** presented inhibition of elastase release, with inhibitory percentages that ranged from 22.1 ± 5.4 to 32.3 ± 6.8% at 10 μM. Inflammation is a defense mechanism response to bacteria, virus, wound or other various environmental factors resulting in injury. It is also a first response of the immune system to infection and stimulation. In response to diverse stimuli, activated neutrophils secrete a series of cytotoxins, such as superoxide anion and elastase [67]. Therefore, inhibition of superoxide anion production and elastase release in infected tissues and organs could directly modulate neutrophil pro-inflammatory responses. Therefore, the crude extract and purified constituents of *P. pterocarpum* have potential to be developed as new anti-inflammatory lead drugs or health food ingredients. In comparison, **39** and **41** enhanced the elastase release in CB-priming human neutrophils with values of 73.0 ± 9.8 and 86.8 ± 3.0 % at 10 μM. It had been reported that an increasing elastase release effect promoted the immune response [68,69]. These results are interesting for the further studies related to the bioactivity and mechanism.

## 3. Materials and Methods

### 3.1. General

All the chemicals, unless specifically indicated otherwise, were bought from Merck KGaA (Darmstadt, Germany). The melting points, optical rotations, UV and IR spectra were recorded on an MP-S3 micromelting point apparatus (Yanagimoto, Kyoto, Japan), a P-2000 digital polarimeter (Jasco, Tokyo, Japan), a U-0080D diode array spectrophotometer (Hitachi, Tokyo, Japan), and a FT-IR Spectrum RX1 spectrophotometer (PerkinElmer, Waltham, MA, USA), respectively. The ESI-MS and HR-ESI-MS spectra were obtained on a Bruker Daltonics APEX II 30e spectrometer (Bruker, Billerica, MA, USA). ^1^H-, ^13^C-, and all 2D NMR (COSY, NOESY, HMQC, and HMBC) spectra were recorded on Bruker AV-500 and Avance III-400 NMR spectrometers (Bruker, Billerica, MA, USA) with tetramethylsilane as the internal standard using deuterated solvents purchased from Sigma-Aldrich (St. Louis, MO, USA). Chemical shifts are reported in parts per million (ppm, δ). Column chromatography and thin layer chromatography (TLC) were conducted on silica gels (Kieselgel 60, 70–230 mesh and 230–400 mesh) and precoated Kieselgel 60 F 254 plates (Merck KGaA), and the compounds were detected by UV light or 10% (*v*/*v*) H_2_SO_4_/EtOH reagent.

### 3.2. Plant Materials

The leaves of *P. pterocarpum* were collected in Vietnam (August 2009) and the plant material was authenticated by Assoc. Prof. Dr. Tran Huy Thai, Institute of Ecology and Biological Resources, Vietnamese Academy of Science and Technology.

### 3.3. Extraction and Isolation

The leaves of *P. pterocarpum* (dried weight 10.0 kg) were powdered, refluxed with methanol and the combined extracts then concentrated *in vacuo* to give a brownish syrup (1.2 kg). The crude extract was further separated into chloroform (350 g) and water soluble layers (850 g) by partition between chloroform and water.

The chloroform layer was purified on a silica gel column eluted with *n*-hexane and a step gradient of ethyl acetate (300:1 to 1:1) to afford nine fractions as monitored by TLC. Fraction 4 was column chromatographed on silica gel with a step gradient mixture of *n*-hexane and ethyl acetate (100:1 to 1:1) to afford 13 subfractions. Subfraction 4.2 was further purified by preparative TLC eluted with a *n*-hexane and ethyl acetate solvent mixture (100:1) to yield **38** (0.8 mg) and **40** (4.2 mg). Subfraction 4.5 was further resolved on a silica gel column eluted with a step gradient mixture of *n*-hexane and acetone (100:1 to 1:1) to produce eight minor fractions (4.5.1–4.5.8). Minor fraction 4.5.2 was purified with pTLC using *n*-hexane and ethyl acetate (50:1) to yield **37** (7.8 mg). Minor fraction 4.5.4 was isolated with silica gel column chromatography with a mixture of benzene and acetone (300:1) and further recrystallization of the resulting fractions afforded **23** (5.5 mg), a mixture of **26** and **27** (6.5 mg), a mixture of **30** and **31** (4.4 mg), **39** (2.7 mg), and **42** (5.6 mg), respectively. Minor fraction 4.5.7 was performed pTLC purification with a mixture of benzene and acetone (50:1) and produced **41** (4.5 mg). Subfraction 4.6 was isolated by silica gel column chromatography eluted with a mixture of benzene and acetone (300:1) and further recrystallization of the minor fractions produced a mixture of **24** and **25** (15.1 mg), a mixture of **28** and **29** (3.2 mg), and **34** (1.5 mg). Subfraction 4.7 was separated by silica gel column chromatography eluted with a chloroform and acetone solvent mixture (300:1) to yield **12** (2.8 mg) and **35** (1.5 mg). Fraction 5 was further separated by repeated column chromatography over silica gel eluted with *n*-hexane and a step gradient of acetone (200:1 to 1:1) followed by purification of the resulting subfractions by recrystallization to afford **1** (2.2 mg), **2** (1.6 mg), **5** (2.7 mg), **11** (2.2 mg), **18** (2.2 mg), and **33** (0.8 mg). Fraction 6 was subjected to silica gel column chromatography eluted with chloroform and a step gradient of methanol (200:1 to 1:1) to produce 13 subfractions. Subfraction 6.6 was further silica gel column chromatographed with a mixture of *n*-hexane and acetone (20:1) to yield **4** (9.0 mg), **7** (4.3 mg) and **20** (7.8 mg). Fraction 7 was resolved on silica gel column eluted with a step gradient mixture of chloroform and methanol (200:1 to 1:1) to give 17 subfractions. Subfraction 7.10 was further separated by silica column chromatography with a mixture of chloroform and acetone (20:1) to result in **10** (1.5 mg). Subfraction 7.11 was isolated by pTLC with a mixture of chloroform and acetone (20:1) to yield **9** (3.1 mg).

The water soluble layer was applied to a reverse-phase Diaion HP-20 column and eluted with a step gradient of water and methanol (10:0, 7:3, 5:5, 3:7, 0:10) to afford nine fractions. Fraction 4 was further purified with Diaion HP-20 column chromatography eluted with water and a step gradient of methanol (10:0 to 0:10) to afford eight subfractions. Subfraction 4.2 was separated by Diaion HP-20 gel column chromatography as previously described to obtain nine minor fractions. The first minor fraction 4.2.1 was further purified by silica gel column chromatography with a mixture of chloroform and methanol (100:1) to produce **6** (2.3 mg) and **14** (4.5 mg). Minor fraction 4.2.4 was further isolated by repeated column chromatography over silica gel eluted with a step gradient mixture of ethyl acetate and methanol (100:1 to 1:1) to result in **3** (4.0 mg), **16** (2.1 mg), **19** (8.2 mg), and **22** (4.8 mg). Minor fraction 4.2.6 was applied to pTLC with a mixture of ethyl acetate, methanol and water (30:1:0.1) to give **54** (7.2 mg). Fraction 5 was resolved on Diaion HP-20 gel column eluted with water and a step gradient of methanol (10:0 to 0:10) to produce eight subfractions. Subfraction 5.4 was further purified by silica gel column chromatography with a mixture of chloroform and methanol (50:1) to yield six minor fractions. Minor fraction 5.4.1 was subjected to pTLC purification with a mixture of chloroform and methanol (100:1) to obtain **15** (9.1 mg). Minor fraction 5.4.2 was further separated by silica gel column chromatography with a mixture of chloroform, methanol and water (10:1:0.1) to result in **50** (14.1 mg), **51** (2.2 mg), and **52** (4.9 mg). Subfraction 5.5 was separated by repeated column chromatography over silica gel eluted with a step gradient mixture of ethyl acetate and methanol (300:1 to 1:1) followed by recrystallization of the resulting minor fractions to yield **8** (7.4 mg), **17** (1.4 mg), **47** (1.9 mg), and **53** (22.4 mg). Fraction 6 was column chromatographed with Diaion HP-20 gel to give 11 subfractions. Subfraction 6.6 was purified by silica gel column chromatography eluted with a step gradient mixture of chloroform, methanol and water (100:1:0.1 to 1:1:0.1) to produce five minor fractions. Minor fraction 6.6.1 was purified by pTLC with a mixture of chloroform, methanol and water (8:1:0.1) to obtain **44** (2.2 g) and **48** (17.3 mg). Subfraction 6.6.2 was purified by repeated column chromatography over silica gel eluted with a step gradient mixture of ethyl acetate, methanol and water (300:1:0.1 to 1:1:0.1) to yield **36** (3.8 mg), **45** (1.5 mg), **46** (12.6 mg), and **49** (2.8 mg). Subfraction 6.7 was further isolated by silica gel column chromatography eluted with a mixture of chloroform, methanol and water (50:1:0.1) to give six minor fractions. The minor fractions 6.7.4 and 6.7.6 were purified by pTLC with a mixture of chloroform, methanol and water (5:1:0.1) to afford **21** (1.6 mg), **43** (6.3 mg); and **13** (4.2 mg), respectively. Fraction 9 was resolved repeatedly on silica gel column chromatography eluted with a step gradient mixture of ethyl acetate and methanol (300:1 to 1:1) followed by recrystallization of the resulting fractions to obtain **32** (1.5 mg).

#### 3.3.1. Peltopterin A (**1**)

White powder, m.p. 116–118 °C (CHCl_3_); [α]_D_^25^ = −73 (*c* 0.1, CHCl_3_); IR (KBr) *ν*_max_ 3430, 2949, 2924, 1709, 1469, 1299, 1239, 1219, 1092, 1026 cm^−1^; ESI-MS (*rel. int. %*) *m*/*z* 221 ([M + Na]^+^, 100); HR-ESI-MS *m*/*z* 221.1150 [M + Na]^+^ (calcd for C_11_H_18_NaO_3_, 221.1154); ^1^H-NMR (CDCl_3_, 400 MHz) δ 4.34 (1H, dd, *J* = 11.1, 4.8 Hz, H-9b), 3.88 (1H, m, H-9a), 3.88 (1H, FWHM = 23.2 Hz, H-3), 2.68 (1H, dd, *J* = 18.1, 5.6 Hz, H-7), 2.20 (1H, dd, *J* = 18.1, 12.2 Hz, H-7), 1.99 (1H, m, H-4), 1.82 (2H, m, H-2, 5), 1.38 (1H, ddd, *J* = 12.2, 12.1, 5.6 Hz, H-6), 1.21 (1H, dd, *J* = 12.1, 6.7 Hz, H-2), 0.95 (3H, s, CH_3_-10), 0.89 (1H, m, H-4), 0.87 (3H, s, CH_3_-11); ^13^C-NMR (CDCl_3_, 100 MHz) δ 171.2 (C-8), 73.8 (C-9), 65.8 (C-3), 50.0 (C-2), 44.2 (C-6), 36.8 (C-4), 34.0 (C-1), 32.3 (C-5), 30.8 (C-7), 29.2 (C-10), 20.2 (C-11).

#### 3.3.2. Peltopterin B (**2**)

Colorless syrup, [α]_D_^25^ = −29 (*c* 0.4, CH_3_OH); UV (MeOH) λ_max_ (log ε) 232 (4.00) nm; IR (KBr) *ν*_max_ 3417, 2963, 1938, 1667, 1455, 1366, 1243, 1157, 1040, 955, 820 cm^−1^; HR-ESI-MS *m*/*z* 241.1434 ([M + H]^+^) (calcd for C_13_H_21_O_4_: 241.1440); ^1^H-NMR (CDCl_3_, 500 MHz) δ 5.84 (1H, s, H-7), 4.33 (1H, dddd, *J* = 11.5, 11.5, 4.0, 4.0 Hz, H-3), 2.29 (1H, ddd, *J* = 11.5, 4.0, 2.5 Hz, H-4_eq_), 2.17 (3H, s, CH_3_-10), 1.98 (1H, ddd, *J* = 11.5, 4.0, 2.5 Hz, H-2_eq_), 1.43 (1H, m, H-4_ax_), 1.42 (3H, s, CH_3_-13), 1.37 (3H, s, CH_3_-11), 1.36 (1H, m, H-2_ax_), 1.15 (3H, s, CH_3_-12); ^13^C-NMR (CDCl_3_, 125 MHz) δ 209.8 (C-8), 198.5 (C-9), 118.8 (C-6), 100.9 (C-7), 72.5 (C-5), 63.4 (C-3), 49.1 (C-2), 48.8 (C-4), 36.3 (C-1), 31.8 (C-12), 31.1 (C-13), 29.2 (C-11), 26.5 (C-10).

#### 3.3.3. Quercetin-3-*O*-[α-l-rhamnopyranosyl(1→3)]-β-d-glucopyranoside (**51**)

Yellow powder; ^1^H-NMR (CD_3_OD, 500 MHz) δ 7.69 (1H, d, *J* = 2.0 Hz, H-2′), 7.58 (1H, dd, *J* = 8.5, 2.0 Hz, H-5′), 6.87 (1H, d, *J* = 8.5 Hz, H-6′), 6.35 (1H, d, *J* = 2.0 Hz, H-8), 6.17 (1H, d, *J* = 2.0 Hz, H-6), 5.74 (1H, d, *J* = 7.5 Hz, H-1″), 5.21 (1H, d, *J* = 1.5 Hz, H-1‴), 4.02 (1H, dd, *J* = 10.0, 6.5 Hz, H-5‴), 3.99 (1H, dd, *J* = 3.5, 1.5 Hz, H-2‴), 3.96 (1H, dd, *J* = 10.0, 8.0 Hz, H-3″), 3.85 (1H, m, H-4″), 3.78 (1H, dd, *J* = 10.0, 3.5 Hz, H-3‴), 3.71 (1H, dd, *J* = 8.0, 7.5 Hz, H-2″), 3.65 (1H, dd, *J* = 11.5, 6.0 Hz, H-6″b), 3.61 (1H, dd, *J* = 11.5, 6.5 Hz, H-6″a), 3.49 (1H, dd, *J* = 6.5, 6.0 Hz, H-5″), 3.34 (1H, m, H-4‴), 0.93 (3H, d, *J* = 6.5 Hz, CH_3_-6‴); ^13^C-NMR (CD_3_OD, 125 MHz) δ 179.4 (C-4), 168.8 (C-7), 163.2 (C-5), 158.4 (C-9), 158.1 (C-2), 149.6 (C-4′), 145.9 (C-3′), 134.6 (C-3), 123.4 (C-6′), 123.0 (C-1′), 117.3 (C-2′), 116.1 (C-5′), 105.8 (C-10), 102.6 (C-1‴), 100.8 (C-1″), 99.9 (C-6), 94.6 (C-8), 77.6 (C-3″), 77.1 (C-5″), 75.8 (C-2″), 74.1 (C-4‴), 72.4 (C-3‴), 72.3 (C-2‴), 70.9 (C-4″), 69.9 (C-5‴), 62.1 (C-6″), 17.4 (C-6‴).

#### 3.3.4. Quercetin 3-*O*-[α-l-rhamnopyranosyl(1→2)]-β-d-xylopyranoside (**52**)

Yellow powder; ^1^H-NMR (CD_3_OD, 500 MHz) δ 7.59 (1H, dd, *J* = 8.5, 2.5 Hz, H-6′), 7.58 (1H, d, *J* = 2.5 Hz, H-2′), 6.87 (1H, d, *J* = 8.5 Hz, H-5′), 6.34 (1H, d, *J* = 2.0 Hz, H-8), 6.16 (1H, d, *J* = 2.0 Hz, H-6), 5.59 (1H, d, *J* = 7.0 Hz, H-1″), 5.02 (1H, d, *J* = 1.0 Hz, H-1‴), 4.08 (1H, qd, *J* = 10.0, 6.0 Hz, H-5‴), 4.00 (1H, dd, *J* = 3.5, 1.0 Hz, H-2‴), 3.79 (1H, dd, *J* = 9.5, 3.5 Hz, H-3‴), 3.75 (1H, dd, *J* = 12.5, 4.5 Hz, H-5″b), 3.70 (1H, dd, *J* = 9.0, 7.0 Hz, H-2″), 3.51 (1H, m, H-4″), 3.51 (1H, m, H-3″), 3.36 (1H, dd, *J* = 10.0, 9.5 Hz, H-4‴), 3.10 (1H, dd, *J* = 12.5, 9.0 Hz, H-5″a), 1.07 (3H, d, *J* = 6.0 Hz, CH_3_-6‴); ^13^C-NMR (CD_3_OD, 125 MHz) δ 179.1 (C-4), 167.5 (C-7), 163.1 (C-5), 158.5 (C-9), 158.4 (C-2), 149.8 (C-4′), 146.2 (C-3′), 134.4 (C-3), 123.3 (C-6′), 123.2 (C-1′), 117.0 (C-2′), 116.0 (C-5′), 105.4 (C-10), 102.7 (C-1‴), 101.3 (C-1″), 100.3 (C-6), 95.0 (C-8), 79.6 (C-2″), 77.9 (C-3″), 74.1 (C-4‴), 72.4 (C-3‴), 72.3 (C-2‴), 71.5 (C-4″), 70.1 (C-5‴), 67.1 (C-5″), 17.6 (C-6‴).

### 3.4. Anti-Inflammatory Bioactivity Examination

The human neutrophils study (No. 1612200032) was approved by the Chang Gung Memorial Hospital Institutional Review Board (Taoyuan, Taiwan) and was conducted according to the Declaration of Helsinki (2013). The examination for the superoxide anion and elastase release inhibition was based on the superoxide dismutase (SOD)-inhibitable reduction of ferricytochrome c and degranulation of azurophilic granules as reported [67]. The experimental details were attached in the file of Appendix A.

## Figures and Tables

**Figure 1 molecules-24-00240-f001:**
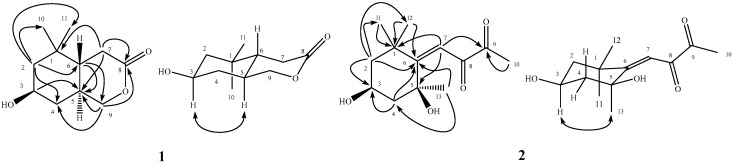
Significant HMBC (→) and NOESY (↔) correlations of **1** and **2**.

**Table 1 molecules-24-00240-t001:** Inhibition of *P. pterocarpum* leave extract and fractions on superoxide anion generation and elastase release in human neutrophils.

Samples	Inhibition Percentage (%) ^a^	
	Superoxide Anion Generation	Elastase Release
Methanol extract	53.4 ± 4.3 ***	112.3 ± 5.0 ***
Chloroform fraction	60.7 ± 5.9 ***	113.6 ± 5.9 ***
Water fraction	49.4 ± 0.7 ***	50.8 ± 5.0 ***

^a^ Percentage of inhibition (Inh %) at 10 μg/mL concentration. Results are presented as mean ± S.E.M. (*n* = 3). *** *p* < 0.001 compared with the control value.

**Table 2 molecules-24-00240-t002:** Superoxide anion and elastase inhibitory effects of isolated compounds in human neutrophils.

Compound	Superoxide Anion	Elastase
	Inh % ^a^	Inh % ^a^
**1**	7.0 ± 3.4	−2.1 ± 3.4
**2**	0.9 ± 2.2	1.5 ± 3.1
**3**	2.5 ± 1.6	−2.1 ± 3.1
**4**	5.5 ± 0.2 ***	1.7 ± 3.5
**5**	4.8 ± 1.4 *	8.3 ± 1.3
**6**	5.4 ± 0.5 ***	6.5 ± 1.3 **
**7**	7.2 ± 2.5 *	3.9 ± 4.1
**8**	3.0 ± 0.6 **	5.3 ± 5.6
**19**	2.2 ± 0.8	7.5 ± 4.1
**20**	10.6 ± 2.6 *	1.7 ± 3.6
**21**	9.2 ± 0.1 ***	9.3 ± 2.9 *
**36**	10.4 ± 6.9	3.0 ± 2.3
**39**	14.0 ± 0.1 ***	– ^b^
**40**	13.4 ± 2.5 **	24.9 ± 3.1 **
**41**	17.1 ± 2.3 **	– ^b^
**43**	42.3 ± 4.3 ***	22.1 ± 6.8 *
**44**	48.5 ± 1.0 ***	12.6 ± 4.0 *
**45**	20.4 ± 4.2 **	14.6 ± 5.9
**46**	−6.8 ± 3.0	−3.5 ± 4.2
**47**	45.7 ± 0.5 ***	22.1 ± 5.4 *
**48**	44.2 ± 4.4 ***	25.5 ± 7.6 *
**49**	−9.1 ± 7.5	17.6 ± 4.4 *
**50**	10.6 ± 1.1 ***	13.2 ± 2.7 **
**51**	46.4 ± 2.7 ***	15.8 ± 3.0 **
**52**	43.7 ± 4.9 ***	32.3 ± 6.8 **
**53**	6.8 ± 2.3 *	17.0 ± 4.9 *
**54**	6.9 ± 1.8 *	18.2 ± 2.9 **

^a^ Percentage of inhibition (Inh %) at 10 μM concentration. Results are presented as mean ± S.E.M. (*n* = 3). * *p* < 0.05, ** *p* < 0.01, *** *p* < 0.001 compared with the control (DMSO). ^b^ The enhance of elastase release was observed at tested concentration.

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
