# Peer review of "Chemical Constituents of the Leaves of *Peltophorum pterocarpum* and Their Bioactivity"

_molecules, 2019, doi:10.3390/molecules24020240_

Round 1
Reviewer 1 Report
Comment for the authors: molecules- 421028
The subject matter of this manuscript is within the aims of the journal. The manuscript deals with chemical structures and their inhibitory effects of superoxide anion generation or elastase release of isolated compounds from Peltophorum pterocarpum (DC.) Backer ex K. Heyne. Some papers on isolated compounds from P. pterocarpum (Raj et al., 2013, Medicinal chemistry research 22, 3823-3830; Jain et al., 2012, Der. Pharm. Chem 4, 2073-2079) have been published. I believe that the studying about the active compounds from P. pterocarpum is important because the isolated compounds are included some potential inhibitors of superoxide anion generation or elastase release, these isolated compounds were not included any new compounds, though. I'm not expert about the English style, because this is not my mother-language, anyway the text seems to me clear, fluent and easy to understand.
Unfortunately, introduction, results, materials and methods, and tables in the manuscript is kind of poor. The manuscript isn’t provided aims of the manuscript, testing superoxide anion generation or elastase release inhibition capacities of the isolated compounds in the introduction section. The manuscript didn’t provide discussion section. Suitable statistical analysis wasn’t used. Therefore, I would recommend it for publication after being revised.
I hope that my comment is useful for the improvement of the article.
Minor comments are noted as follows;
1. Introduction
Aims of the manuscript, testing superoxide anion generation or elastase release inhibition capacities of the isolated compounds should be shown.
2. Results
The manuscript didn’t provide discussion section. I think results section means results and discussion section.
3. Materials and Methods
A figure of fractionation can be shown because the fractionation was kind of complicated.
Line 207
“The leaves of P. pterocarpum (10.0 kg)”
Was the weight dry weight? Please show details.
3.4. Anti-inflammatory Bioactivity Examination
More detail information of the bioactivity examination should be shown. You can show the information in supporting information.
3.4.3 Statistical Analysis
Line 329 “Calculations of 50 % inhibitory concentrations (IC50) were computer-assisted (PHARM/PCS v.4.2)”
I cannot find IC50 values in the manuscript.
I think you should choice multiple comparison procedure Dunnett's test or Steel's test because the statistical analysis to compare each group with the control group in the manuscript. Was normality of distribution evaluated? If the data set was well-modeled by a normal distribution, Dunnett's test is better choice. If the data set wasn’t well-modeled by a normal distribution, Steel's test is better choice.
Table 2
What does “-” mean? You should explain in the caption.
Author Response
The subject matter of this manuscript is within the aims of the journal. The manuscript deals with chemical structures and their inhibitory effects of superoxide anion generation or elastase release of isolated compounds from Peltophorum pterocarpum (DC.) Backer ex K. Heyne. Some papers on isolated compounds from P. pterocarpum (Raj et al., 2013, Medicinal chemistry research 22, 3823-3830; Jain et al., 2012, Der. Pharm. Chem 4, 2073-2079) have been published. I believe that the studying about the active compounds from P. pterocarpum is important because the isolated compounds are included some potential inhibitors of superoxide anion generation or elastase release, these isolated compounds were not included any new compounds, though. I'm not expert about the English style, because this is not my mother-language, anyway the text seems to me clear, fluent and easy to understand.
Unfortunately, introduction, results, materials and methods, and tables in the manuscript is kind of poor. The manuscript isn’t provided aims of the manuscript, testing superoxide anion generation or elastase release inhibition capacities of the isolated compounds in the introduction section. The manuscript didn’t provide discussion section. Suitable statistical analysis wasn’t used. Therefore, I would recommend it for publication after being revised.
Response: Thank you for your comments. We revised the text according to your suggestions listed below.
Minor comments are noted as follows.
1. Introduction: Aims of the manuscript, testing superoxide anion generation or elastase release inhibition capacities of the isolated compounds should be shown.
Response: We had added the sentence regarding the aim of this study in the introduction section.
2. Results: The manuscript didn’t provide discussion section. I think results section means results and discussion section.
Response: We revised the section 2 as “Results and Discussion” as reviewer suggested.
3. Materials and Methods: A figure of fractionation can be shown because the fractionation was kind of complicated.
Response: Thank you for your suggestion. We provided the Extraction and isolation schemes in the Supporting Information file (S1).
4. Line 207, “The leaves of P. pterocarpum (10.0 kg)”. Was the weight dry weight? Please show details.
Response: We showed the details in the extraction and isolation section.
5. Anti-inflammatory Bioactivity Examination: More detail information of the bioactivity examination should be shown. You can show the information in supporting information.
Response: Thank you for your comment. We provided the complete anti-inflammatory bioactivity experimental procedures in the Supporting Information file (S2).
6. Statistical Analysis: Line 329 “Calculations of 50 % inhibitory concentrations (IC50) were computer-assisted (PHARM/PCS v.4.2)”. I cannot find IC50 values in the manuscript. I think you should choice multiple comparison procedure Dunnett's test or Steel's test because the statistical analysis to compare each group with the control group in the manuscript. Was normality of distribution evaluated? If the data set was well-modeled by a normal distribution, Dunnett's test is better choice. If the data set wasn’t well-modeled by a normal distribution, Steel's test is better choice.
Response: Thank you for your comment. We had removed these sentences from the text and provided the complete anti-inflammatory bioactivity experimental procedures in the Supporting Information file (S2). In addition, the calculations of 50 % inhibitory concentrations (IC50) were deleted since no examined compounds exceeded 50 % inhibition in the tested concentrations.
7. Table 2: What does “-” mean? You should explain in the caption.
Response: In Table 2, “-” means the enhance of elastase release and this was added in the footnote.
Reviewer 2 Report
This manuscript entitled "Chemical Constituents of the Leaves of Peltophorum pterocarpum and Their Bioactivity" describes the three new compounds isolated from extract of peltophorum pterocarpum. The authors isolated fifty-one compounds and for three new and two described elsevier discussed the 1H- and 13C NMR.
The chemical part does not raise any doubts, but the biological part raises some doubts. The authors should emphasize why only the activity of isolated compounds as inhibitory effect of superoxide anion generation and elastase release were examined.
I think that after small correction this manuscript should be accepted for publication in Molecules.
Author Response
This manuscript entitled "Chemical Constituents of the Leaves of Peltophorum pterocarpum and Their Bioactivity" describes the three new compounds isolated from extract of peltophorum pterocarpum. The authors isolated fifty-one compounds and for three new and two described elsevier discussed the 1H- and 13C NMR.
The chemical part does not raise any doubts, but the biological part raises some doubts. The authors should emphasize why only the activity of isolated compounds as inhibitory effect of superoxide anion generation and elastase release were examined.
I think that after small correction this manuscript should be accepted for publication in Molecules.
Response: Thank you for your comments. We had added the sentence regarding the aim of this study in the introduction section. Since in our preliminary bioassay, the methanol extract and fractions of leaves of P. pterocarpum displayed significant inhibition of superoxide anion generation and elastase release, and thereby we elucidated the chemical compositions and subjected the purified compounds to the inhibition tests of superoxide anion generation and elastase release for discovery of new anti-inflammatory leads from natural sources. We hoped that these changes could emphasize our aim of this study.
Reviewer 3 Report
This manuscript presents the structure elucidation and the biological activity of new terpenoids from the leaves of Peltophorum pterocarpum. HRMS data is one of the most important evidence for the structure determination of new compounds. However, HRESIMS data are not provided for compounds 2and 3. The authors should provide HRESIMS dataof compounds 2and 3in Experimental Section, as well as the MS spectra of compounds 1-3(in Supporting Information).
The manuscript needs corrections as follows.
(1) Page 3, line 89: "the pseudomolecular ion" should be revised to "a sodium adduct ion".
(2) Page 3, line 92: "conjugated"should be removed.
(3) Page 3, lines 108-110: The authors mentioned “the correlations between H-3 and H-5 in the nuclear Overhauser effect spectroscopy (NOESY) established its relative stereochemistry configuration (Figure 2).”. However, since H-5 signal (1.82 ppm) of compound 1 overlapped with that of H-2 (1.82 ppm), the NOESY correlation may be attributed to that between H-2 and H-3. Therefore, the relative configurations of C-5 and C-6 have not been determined yet. Compound 1 may have a cis-junction ring system. The authors should demonstrate the relative configurations of C-5 and C-6.
(4) Page 3, lines 126-127: The authors mentioned “The NOESY spectrum showed NOE correlations between axial H-3 and H-13 determined the OH-5 as equatorial (Figure 2).”. However, since H-13 signal (1.42 ppm) of compound 2 overlapped with that of H-4a (1.43 ppm), the NOESY correlation may be attributed to that between H-4a and H-3. Therefore, the relative configuration of C-5 has not been determined yet. The authors should demonstrate the relative configuration of C-5 of compound 2.
(5) Page 4, lines 143-146: There is no evidence for the relative configurations of C-1, C-5, and C-6 of compound 3. The authors should demonstrate the relative configurations of C-1, C-5, and C-6 of compound 3.
(6) Pages 3-4: The authors should mention the absolute configurations of compounds 1-3.
(7) Page 8, lines 323-326: The authors should describe bioassay methods in more detail.
(8) Page 8, line 329: The authors mentioned calculations of IC50. If IC50 for each compound was determined, the authors should describe the IC50 values in Results.
Author Response
This manuscript presents the structure elucidation and the biological activity of new terpenoids from the leaves of Peltophorum pterocarpum. HRMS data is one of the most important evidence for the structure determination of new compounds. However, HRESIMS data are not provided for compounds 2 and 3. The authors should provide HRESIMS data of compounds 2 and 3 in Experimental Section, as well as the MS spectra of compounds 1-3 (in Supporting Information).
Response: Thank you for your comment. We had provided the HRESIMS data of compounds 2 and 3 in Experimental Section, as well as the MS/HRMS spectra of compounds 1-3 in Supporting Information.
The manuscript needs corrections as follows.
1. Page 3, line 89: "the pseudomolecular ion" should be revised to "a sodium adduct ion".
Response: This was revised.
2. Page 3, line 92: "conjugated" should be removed.
Response: It was removed.
3. Page 3, lines 108-110: The authors mentioned “the correlations between H-3 and H-5 in the nuclear Overhauser effect spectroscopy (NOESY) established its relative stereochemistry configuration (Figure 2).”. However, since H-5 signal (1.82 ppm) of compound 1 overlapped with that of H-2 (1.82 ppm), the NOESY correlation may be attributed to that between H-2 and H-3. Therefore, the relative configurations of C-5 and C-6 have not been determined yet. Compound 1 may have a cis-junction ring system. The authors should demonstrate the relative configurations of C-5 and C-6.
Response: Thank you for your comment. Although the NOESY correlation mentioned above may be attributed to H-2 and H-3 rather than H-3 and H-5, the cis-ring junction system would result in the significant NOESY correlations between H-5 and H-6. However, this correlation was not observed. Therefore, Compound 1 may have a trans-junction ring system and the stereochemistry should be indicated as shown in Figure 1.
4. Page 3, lines 126-127: The authors mentioned “The NOESY spectrum showed NOE correlations between axial H-3 and H-13 determined the OH-5 as equatorial (Figure 2).”. However, since H-13 signal (1.42 ppm) of compound 2 overlapped with that of H-4a (1.43 ppm), the NOESY correlation may be attributed to that between H-4a and H-3. Therefore, the relative configuration of C-5 has not been determined yet. The authors should demonstrate the relative configuration of C-5 of compound 2.
Response: Thank you for your comment. We checked and expanded the NOESY spectrum carefully to confirm the position of H-13 and H-4eq. The proton signal at δ 2.29 (1H, ddd, J = 11.5, 4.0, 2.5 Hz, H-4ax) was determined as axial due to its coupling constant (J = 11.5 Hz). In addition, H-4ax showed NOE correlation with CH3-13 (See Supporting Information). Therefore, the relative configuration of 2 was corrected as shown and Figures 1 and 2 were revised correspondingly.
5. Page 4, lines 143-146: There is no evidence for the relative configurations of C-1, C-5, and C-6 of compound 3. The authors should demonstrate the relative configurations of C-1, C-5, and C-6 of compound 3.
Response: Thank you for your comment and we revised the elucidation of compound 3. In its 1H-NMR spectrum, the coupling constant between H-3 and H-4ax indicated H-3 as axial. The NOESY correlations observed for H-3/H-2eq, H-3/H-4eq, H-4ax/CH3-14, H-7/CH3-14, and H-7/CH3-13 established the relative configurations C-1, C-3, C-5, and C-6 as shown in the text.
6. Pages 3-4: The authors should mention the absolute configurations of compounds 1-3.
Response: Thank you for your suggestion. The absolute configurations of compounds 1-3 may be determined through some modification experiments. However, we are sorry that we do not have enough amount to prepare the derivatives and therefore we could not determine their absolute stereochemistry.
7. Page 8, lines 323-326: The authors should describe bioassay methods in more detail.
Response: Thank you for your comment. We had provided the complete anti-inflammatory bioactivity experimental procedures in the Supporting Information file (S2).
8. Page 8, line 329: The authors mentioned calculations of IC50. If IC50 for each compound was determined, the authors should describe the IC50 values in Results.
Response: The calculations of 50 % inhibitory concentrations (IC50) were deleted since no examined compounds exceeded 50 % inhibition in the tested concentrations.
Round 2
Reviewer 3 Report
The manuscript-v2 was revised insufficiently.
(1) 1H NMR assignments for compound 2: The proton signals at delta 1.98 and 2.29 should be attributable to H-2eq and H-4eq, respectively, because the large coupling constant of 11.5 Hz of the signals is due to geminal coupling. The relative configuration of C-5 remains to be determined as well as the geometry of the double bond at C-6. The authors should measure its NOESY spectrum in the other solvent (e.g., C6D6, C5D5N) to determine the relative configuration or remove the stereochemistry of compound 2 from the text.
(2) Structure of compound 3: The 13C chemical shift of C5 (delta 90.1) suggests the presence of a lactone ring, which is supported by the HRMS data (m/z 295.1183 corresponding to [M-H]-). The 13C chemical shifts of compound 3 are similar to those of rel-5-(3S,8S-dihydroxy-1R,5S-dimethyl-7-oxa-6-oxobicyclo[3,2,1]oct-8-yl)-3-methyl-2Z,4E-pentadienoic acid described in the literature (H. Kikuzaki et al., J. Agric. Food Chem., 2004, 52, 344). The authors should reconsider the structure.
(3) The authors should mention the absolute configurations in the text e.g., the absolute configurations of the new compounds remain to be determined.
Author Response
(1) 1H NMR assignments for compound 2: The proton signals at delta 1.98 and 2.29 should be attributable to H-2eq and H-4eq, respectively, because the large coupling constant of 11.5 Hz of the signals is due to geminal coupling. The relative configuration of C-5 remains to be determined as well as the geometry of the double bond at C-6. The authors should measure its NOESY spectrum in the other solvent (e.g., C6D6, C5D5N) to determine the relative configuration or remove the stereochemistry of compound 2 from the text.
Response: Thank you for your comment. We had recorded again the 1H NMR and NOESY spectra of 2 in C6D6 and the results were attached in the Supplementary Materials. In this NOESY spectrum, NOE correlation between H-3 and CH3-13 was clearly found. Therefore, we revised again the stereochemistry at C-5. In addition, the double bond at C-6 should adapt a half-chair form rather than a standard chair form. Conclusively, we revised the structure as shown in the text.
(2) Structure of compound 3: The 13C chemical shift of C5 (delta 90.1) suggests the presence of a lactone ring, which is supported by the HRMS data (m/z 295.1183 corresponding to [M-H]-). The 13C chemical shifts of compound 3 are similar to those of rel-5-(3S,8S-dihydroxy-1R,5S-dimethyl-7-oxa-6-oxobicyclo[3,2,1]oct-8-yl)-3-methyl-2Z,4E-pentadienoic acid described in the literature (H. Kikuzaki et al., J. Agric. Food Chem., 2004, 52, 344). The authors should reconsider the structure.
Response: Thank you for your comment. After careful comparison of our spectra with those in the literature, we hope that we could revise the new compound 3 as a known compound mentioned above by the reviewer. We are sorry to make this mistake and the references' numbers were changed accordingly.
(3) The authors should mention the absolute configurations in the text e.g., the absolute configurations of the new compounds remain to be determined.
Response: Thank you for your suggestion. We had inserted this sentence in the text (line 131-132) as you suggested.